# Boosting Sustainability through Digital Transformation's Domains and Resilience

**Reihaneh Hajishirzi** [1,*] **, Carlos J. Costa** [1] **and Manuela Aparicio** [2]

1    Advance/ISEG (Lisbon School of Economics & Management), Universidade de Lisboa,
     1200-109 Lisbon, Portugal; cjcosta@iseg.ulisboa.pt
2    NOVA Information Management School (NOVA IMS), Universidade Nova de Lisboa,
     1070-312 Lisbon, Portugal; manuela.aparicio@novaims.unl.pt
*    Correspondence: reihaneh.hajishirzi@aln.iseg.ulisboa.pt

**Abstract:** Sustainability is a must for all businesses in all industries. It can boost company image and productivity while being aligned with customer needs. On the other hand, digital transformation (DT) is vital for business environments, and organizations need to be resilient in the face of crises such as COVID-19. The main objective of our study is to figure out how DT and organizational resilience might help businesses become more sustainable. This study presents a model that explains social, environmental, and economic sustainability considering the domains of DT and organizational resilience. Our model is evaluated on the data gathered by 208 c-suite leaders from various Iranian companies. The model was empirically validated through a quantitative method of Partial Least Squares/Structural Equation Modeling (PLS/SEM) technique. The findings reveal that the five studied factors have substantial impact on the sustainability of Iranian organizations including data-driven, business process innovation, customer engagement, organizational resilience, and competitive advantages.

**Keywords:** business process innovation; sustainability; competitive advantage; digital transformation; customer engagement

## 1. Introduction

In today's business world, Digital Transformation (DT) and sustainability are two major market factors for organizations [1]. Boston consulting group (BCG) [2] derives a new mindset called "technology ecoadvantage", which means utilizing cutting edge technologies and digitized operations to develop lucrative solutions that bring sustainability. There are a lot of advantages for firms if they include environmental, social and economic sustainability while adopting their digital transformation strategies [3,4].

According to Rogers [5], DT has five domains: data, innovation, customer, competition and value. He believes that data is created quickly with exponential growth [6,7], but analyzing and generating meaningful information is challenging. Organizations that can adopt a data-driven strategy will create more value [5,8]. Moreover, digital technologies enable rapid innovation in the processes and products, which help firms be leaders in the market [5,9]. In addition, digitalization changes customer experiences through customer engagement [5,10] and creates success for organizations in a competitive environment by focusing on platform business models and competitive advantages [5,11]. On the other hand, organizational resilience is a dynamic capability for responding in times of disruption and crisis [12–14]. In the face of the COVID-19 pandemic, organizations need to strengthen their resilience by engaging with stakeholders, promoting virtual work, and driving customer communication [15]. The firms that can predict the crisis have more social and economic sustainability [16].

This study targets Iranian organizations, where DT and sustainability appear in the beginning phases of implementation [17]. The literature emphasizes that there is still a need

for more research for Iranian companies which are located in an important geographical area but not enough studies have been done in this regard [18–20]. In addition, more research is still required to better understand of the effects of DT on sustainability [21,22]. Accordingly, the main objective of our research is to identify how can digital transformation process enable a transition to more sustainable companies. Our specific objectives are:

- RQ1: What is the impact of digital transformation's domains on sustainability?
- RQ2: What is the impact of organizational resilience on sustainability?

To achieve these objectives, we propose a new theoretical model. We conduct an empirical study to understand the impacts of DT's domains and organizational resilience on social, environmental, and economic sustainability in Iranian companies. We gather and analyze data from 208 actual firms to verify this approach.

The study results show that business process innovation, customer engagement, and competitive advantage significantly affect sustainability (social, environmental, and economic). This research contributes to the current literature on DT, sustainability, and organizational resilience by focusing on data, innovation, customer engagement and competitive advantage. It also contributes to improve understanding of organizational sustainability.

The following is a breakdown of the paper's structure. Section 2 contains the literature review. In part 3, the conceptual model and hypotheses are provided, followed by a description of how the empirical investigation was carried out in Section 4. The results and comments are summarized in Section 5. The conclusion is offered in the final Section 6.

## 2. The Current State of the Art

According to Brundtland World Commission Report [23], sustainability is the development that provides the current demands without harming future generation's capacity to fulfill their needs. Sustainability has three aspects: social, environmental, and economic [24]. Social sustainability is concerned about the impacts that organizations have on the available capacity of non-financial capital [25]. Environmental sustainability is a state of stability, resilience, and coherence that permits humans to achieve their demands while not surpassing the ability of their supporting ecosystem to renew the services required to meet those needs, nor reducing biological variety via activities [26]. Finally, economic sustainability pertains to the company's growth, development, and productivity [27]. There are several benefits for businesses which include environmental, social, and economic sustainability in their DT strategy [3,4]. Previous research shows digitization improves environmental sustainability [28]. The other study provides a literature review over DT and sustainability and presents a framework with four key areas including: pollution control, waste management, sustainable production, and urban sustainability [21]. Other researchers study how the big companies have tackled sustainable development, covering a variety of challenges in the digital transformation topic [29].

The resource-based view theory in information system literature, describes that organizations can achieve competitive advantages by increasing rare, valuable imitable, and substitutable firm's capabilities [30]. Previous study finds that IT capabilities are organizational capabilities and it can increase competitive advantages [31]. According to Rogers [5], in digital age companies should build platforms—instead of just focusing on product developments—to gain competitive advantage. Platform theory states that companies can create value in network of other partners and rivals [32]. It is also important to study the behavior of competitors and assess the value of the new entrants in the market [33]. Another line of research shows that innovation and technological capabilities have an impact on sustainable competitive advantage [34].

Rogers [5] describes innovation as adding value to a company product, service or process. Based on the theory of disruption [35], in complex and high-cost markets, companies, by applying innovation, can transform their market with convenient, cost-effective and transparent solutions. In the digital era, startups can use new technologies like artificial intelligence, blockchain, Internet of things, cloud computing, and additive manufacturing [36,37] and create new business models which make incumbents face digital

disruption [38]. According to business model innovation theory [39], if incumbents want to survive and create lean value, at first, they should understand the current business model and focus on people and their relationships and behaviors. Business process innovation deploys a novel and considerably enhanced production process and distribution mechanism [40]. Another study proposes a framework of dynamic capabilities that organizations can use for developing digital process innovation [41].

The resilience topic has been recently studied in management scholar [42,43] mainly during the COVID-19 crisis. Organizational resilience refers to a company's response to being destroyed, and it emphasizes the ability to recover and flourish in the face of adversity, crisis, or disaster [44]. It emphasizes a company's capability to adapt, expand, and survive in a changing environment [45]. To be resilient, organizations need to change their culture [46] and become more agile. Resilient businesses focus on assessing what type of business they want to be and how they can acquire a competitive edge that will help them achieve it [47].

Table 1 classified the previous research on DT domains and sustainability and determines the domains they use to measure organizational sustainability.

**Table 1.** Previous research on DT and sustainability.

| Study | Description | Methodology | Variables | | | | |
|-------|-------------|-------------|-----------|------------|----------|-------------|------------|
| | | | Data | Innovation | Customer | Competition | Resilience |
| [48] | They propose a framework that shows the impact of customer engagement on company value. | Literature Review | ● | | ● | | |
| [49] | They present a model for applying big data analytics on sustainable customer market. | Quantitative approach | ● | | ● | | |
| [50] | They construct a model for the impact of big data on customer behavior. | Quantitative approach | ● | | ● | | |
| [51] | They present seven future trends, and the first is about changing customer experience, customer involvement, and customer satisfaction. | Literature Review | | ● | ● | | |
| [52] | They propose a model to understand the effect of innovation capability and customer experience on relationships. | Quantitative approach | | ● | ● | | |
| [53] | They study several ICT companies to understand the relationship between customer engagement and business process innovation. | Case Study | | ● | ● | | |
| [54] | They present a framework that shows the impact of customer engagement on competitive advantage and firm performance. | Quantitative approach | | | ● | ● | |
| [55] | They propose a framework that shows the importance of customer engagement on sustainable competitive advantage. | Qualitative approach | | | ● | ● | |
| [56] | This study shows that customer experience could bring competitive advantage and create value. | Qualitative and Quantitative approach | | | ● | ● | |
| [57] | This research shows that customer engagement impacts sustainable development | Quantitative approach | | | ● | | |
| [58] | This article reveals that customer engagement influences corporate social responsibility. | Quantitative approach | | | ● | | |

**Table 1.** *Cont.*

| Study | Description | Methodology | Variables | | | | |
|---|---|---|---|---|---|---|---|
| | | | Data | Innovation | Customer | Competition | Resilience |
| [16] | They present a model that shows the influence of organizational resilience on sustainability. | Quantitative approach | | | | | • |
| [59] | This study constructs a conceptual model to show the relationship between resilience and sustainability. | Literature Review | | | | • | • |
| [60] | This study investigates the competitive advantage of the environmental behavior. | Case Study | | | | • | |
| [61] | This research proposes a model for analyzing the impact of competitive advantage on economic sustainability. | Quantitative approach | | | | • | |
| [34] | This article presents a model for sustainable competitive advantages. | Quantitative approach | | • | | • | |

## 3. Research Model

The aim of this research is to figure out how DT domains influence the sustainability of Iranian companies. The constructions, hypotheses, and theoretical model are all described in this section. To propose our research model, we use digital transformation domains which were introduced by Rogers [5]. Moreover, we use the organizational resilience domain which has become an important construct during the COVID-19 pandemic.

This model is made up of eight different constructs, which are: Data-Driven (Data), Business Process Innovation (Inn), Customer Engagement (Cus), Organizational Resilience (Res), Competitive Advantage (CompetAdv), Economic Sustainability (EcoS), Environmental Sustainability (EnvS), and Social Sustainability (SocS). Table 2 shows the definition of the constructs.

**Table 2.** Construct Definition.

| Construct | Definition | Reference |
|---|---|---|
| Data-Driven (Data) | In today's digital world, data is continuously generated everywhere. The challenge of data is turning it into valuable information. Unstructured data is increasingly usable and practical. The value of data is connecting it across silos. Data is a critical intangible asset for value creation. | [5] |
| Business Process Innovation (Inn) | In today's rapidly changing business environment, organizations should be more responsive and agile to customers' needs. They need to apply innovative solutions and technologies in their processes to reduce costs and improve quality. | [44] |
| Customer Engagement (Cus) | It deals with satisfying consumers by giving more value in comparison to competitors to foster continuous relations based on belief and trust. | [62] |
| Organizational Resilience (Res) | Refers to an organization's response to damage and it highlights the ability to recover and grow under uncertainty, crisis, and emergency. | [63] |
| Competitive Advantage (CompetAdv) | When a company obtains a set of traits that enable it to work better than its competitors, it gains a competitive advantage. The competitive advantage is shown when a firm's actions are more profitable than its competitors or when it beats them in terms of other essential incomes. | [64,65] |
| Economic Sustainability (EcoS) | It relates to the growth, development, and productivity of the company. It means that we should optimize the usage of resources to create long-term sustainable values in our organization. | [27] |

**Table 2.** *Cont.*

| Construct | Definition | Reference |
|---|---|---|
| Environmental Sustainability (EnvS) | It refers to the maintenance of natural capital; for this reason, firms should take care of waste emissions and use renewable and nonrenewable resources very carefully. | [66] |
| Social Sustainability (SocS) | It is a technique for developing sustainable locations that encourage wellness by understanding people's requirements of their living and working places. It relates to the physical and social infrastructures, social amenities, and citizen participation mechanisms. | [67] |

Figure 1 indicates our proposed. It presents that data-driven (Data) and business process innovation (Inn) affects customer engagement (Cus). Customer engagement (Cus) influences social sustainability (SocS), environmental sustainability (EnvS), and competitive advantage (CompetAdv). Organizational resilience (Res) has an impact on competitive advantage (CompetAdv) and economic sustainability (EcoS). In addition, competitive advantage (CompetAdv) affects social, environmental, and economic sustainability. Finally, organizational resilience has a moderating effect on relation of competitive advantage and economical sustainability.

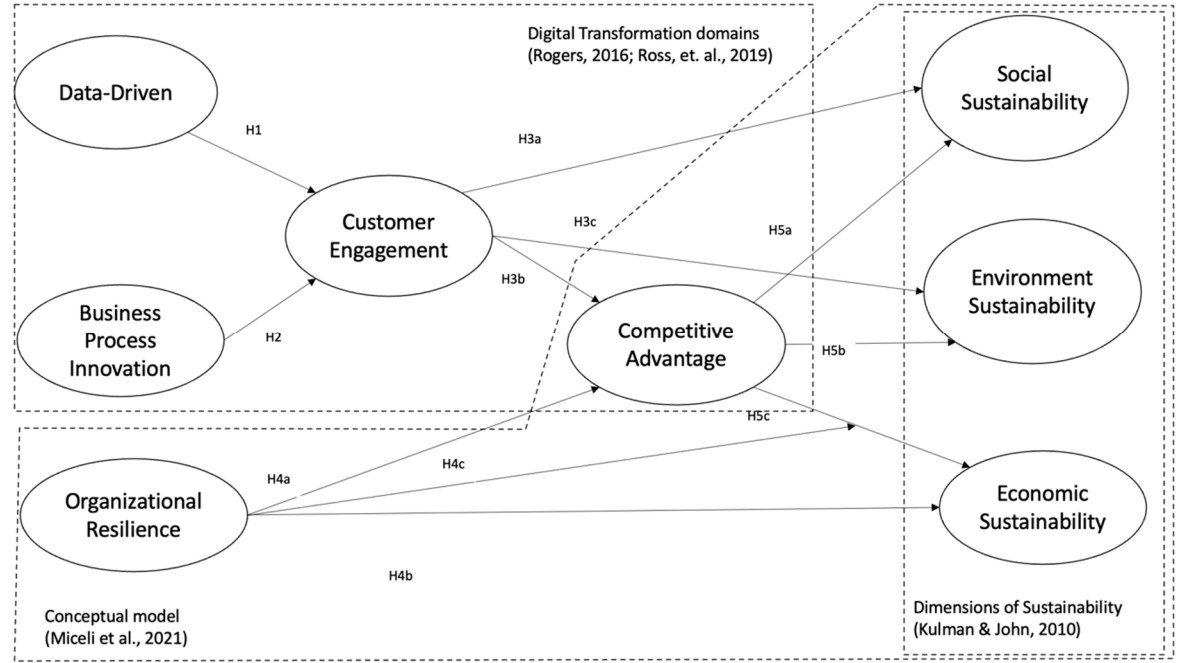

**Figure 1.** Sustainability explained by digital transformation model.

According to OECD [68], the use of data and analytics provide value to customer interactions. Previous study analyzes the impact of big data on consumer purchase intention and understood that it positively impacts the desire of customers on e-commerce [69]. Data mining helps firms find potential customers, define customer segmentation, and improve customer retention [70]. Companies should invest in technologies to successfully manage big data to boost customer engagement [71]. According to Rogers [5], data gathering from customers becomes one of every organization's most valuable assets. Data can help determine which clients demand the most significant attention, and it is utilized to help companies customize their client communications. Even firms can develop data-driven business models that retrieve and sell external data to meet customers' demands [72]. Big data analysis increases customer attention and purchasing behavior [50]. The difficulty of obtaining data from various channels on customer engagement is one of the most critical challenges [48]. Accordingly, the following hypothesis is suggested:

**Hypothesis 1 (H1).** *Data-driven have a positive effect on customer engagement.*

Today's businesses are constantly under competitive pressure [73]. One of the ways that companies can survive is to put customer engagement at the center of a firm's innovation process [74]. There is another perspective that the presence of innovation capabilities influences the loyalty and reputation of the company [52]. Another line of research analyzes the business process innovation in Apple, Google, and Microsoft to understand the critical role of customer engagement in process innovation activities [53]. Other researchers conduct a systematic literature review on digital innovations and business process management. They present seven future trends, and the first is about changing customer experience, customer involvement, and customer satisfaction [51]. Accordingly, the following hypothesis is suggested:

**Hypothesis 2 (H2).** *Business process innovation has a positive effect on customer engagement.*

Previous work shows a framework that shows the impact of customer engagement on competitive advantage and firm performance [54]. Another line of research shows the importance of customer engagement on sustainable competitive advantage [55]. An empirical study on the city transportation domain shows that customer experience leads to competitive advantage and creates value [56]. On the other hand, previous studies show that customer engagement impacts sustainable development [57] and corporate social responsibility [58,75]. Accordingly, the following hypotheses are suggested:

**Hypothesis 3a (H3a).** *Customer engagement has a positive effect on social sustainability.*

**Hypothesis 3b (H3b).** *Customer engagement has a positive effect on competitive advantage.*

**Hypothesis 3c (H3c).** *Customer engagement has a positive effect on environmental sustainability.*

Organizational resilience should be linked to a competitive advantage for a company [76]. Resilient companies are focused on determining what type of firms they want to be and how they may acquire a competitive advantage that will enable them to achieve it [47]. An empirical study shows that organizational resilience positively affects economic sustainability [16]. Researchers believe that organizational resilience must be viewed as a subterm, comparable to the holistic perspective of sustainability [59]. There are also several studies that show a moderating effect of organizational resilience [77–79]. Accordingly, the following hypotheses are suggested:

**Hypothesis 4a (H4a).** *Organizational resilience has a positive effect on competitive advantage.*

**Hypothesis 4b (H4b).** *Organizational resilience has a positive effect on economic sustainability.*

**Hypothesis 4c (H4c).** *Organizational resilience has a moderating effect on a relation of competitive advantage and economical sustainability.*

Several researchers conducted empirical research and found that companies' competitive advantages are one of the most essential benefits of firms tackling sustainability challenges [80]. A new perspective shows that the high-tech companies create high margin businesses and sustainable competitive advantages when they develop imitable resources and capabilities [81]. Moreover, scientists investigate the competitive advantage of environmental behavior at a company level [60]. Another study shows that competitive activities have a positive effect on sustainable competitive advantage and competitive advantage has a positive impact on business performance [61]. The other line of research shows that sustainable advantage has a positive effect on market performance [34]. Accordingly, the following hypotheses are suggested:

**Hypothesis 5a (H5a).** *Competitive advantage has a positive effect on social sustainability.*

**Hypothesis 5b (H5b).** *Competitive advantage has a positive effect on environmental sustainability.*

**Hypothesis 5c (H5c).** *Competitive advantage has a positive effect on economic sustainability.*

## 4. Empirical Study & Results

We develop a study instrument using the measurement model (Appendix A) to survey a random sample of Iranian businesses and organizations. Our measurement model is a questionnaire divided into two sections: (1) questions regarding sample characteristics, (2) questions regarding construct evaluations. On a seven-point scale, respondents may choose their replies (1—Strongly Disagree to 7—Strongly agree).

We use data-driven and business process innovation domains to measure the effect of customer engagement. Further, we use customer engagement and organizational resilience to measure the competitive advantage effect. In addition, we apply customer engagement and competitive advantage to measure social sustainability and environmental sustainability. Finally, we use competitive advantage and organizational resilience to measure economic sustainability.

We apply the two-step PLS/SEM method for data analysis using the SmartPLS 3.0 tool [82,83]. This section describes the structural model outcomes after presenting the measurement model data [84].

### 4.1. Sample Characterization

Our data consists of 208 responses collected from May to November 2021, using our questionnaire which was distributed via Google form. Table 3 shows the characteristics of the respondents. About 89% of respondents are male and 47.59% respondents are between 31 to 40 years old. Respondents include small to big companies in various industries, including manufacturing, services, and construction.

**Table 3.** Descriptive statistics of respondent characteristics.

| Respondent Characteristics | (n = 208) | |
|---|---|---|
| **Gender** | | |
| Female | 23 | 11.05% |
| Male | 185 | 88.95% |
| **Age** | | |
| 18–30 | 30 | 14.42% |
| 31–40 | 99 | 47.59% |
| 41–50 | 55 | 26.45% |
| 51–60 | 20 | 9.62% |
| >60 | 4 | 1.92% |
| **Organization Characteristics** | | |
| **Age of the Organization** | | |
| <2 | 31 | 14.90% |
| 2–5 | 43 | 20.67% |
| 6–10 | 35 | 16.83% |
| 11–20 | 51 | 24.52% |
| >20 | 48 | 23.08% |

**Table 3.** *Cont.*

| Respondent Characteristics | (n = 208) | |
|---|---|---|
| **Industry** | | |
| Charity/not for profit | 0 | 0% |
| Construction/Property | 8 | 3.64% |
| Consumer Packaged Goods | 4 | 1.82% |
| Education | 8 | 3.18% |
| Energy/Mining | 20 | 9.55% |
| Entertainment/media | 4 | 1.82% |
| Financial services | 20 | 9.09% |
| Hospitality/Catering | 0 | 0% |
| IT and technology | 69 | 31.36% |
| Legal | 1 | 0.45% |
| Manufacturing | 26 | 11.82% |
| Pharmaceutical | 10 | 4.54% |
| Private healthcare and services | 4 | 1.82% |
| Professional/Business services | 17 | 7.73% |
| Public sector (incl. local and central government, etc.) | 13 | 5.91% |
| Retail | 4 | 1.82% |
| Telecommunications | 2 | 0.91% |
| Transport, distribution, and logistics | 9 | 4.09% |
| Utilities | 1 | 0.45% |

*4.2. Measurement Model Assessment*

We use the PLS technique to determine if the constructs were trustworthy or not. Table 4 summarizes the measurement model results across different metrics including outer loading, composite reliability, Cronbach's Alpha, and average variance extracted (AVE). The outer loading is about an indicator weight, and it should be over 0.70 [85]. The composite reliability shows the internal consistency in scale items and it should be more than 0.70 [86]. Cronbach's Alpha is another metric of dependability, and if it is more than 0.7, it indicates that the study is internally consistent [87]. The average variance extracted (AVE) indication shows the constructs' convergent validity, and it should be more than 0.5 [88].

**Table 4.** Measurement model results.

| Construct | Items | Outer Loading | Composite Reliability | Cronbach's Alpha | AVE | Discriminant Validity? |
|---|---|---|---|---|---|---|
| Data | Data1 | 0.932 | 0.907 | 0.848 | 0.766 | Yes |
| | Data2 | 0.912 | | | | |
| | Data3 | 0.773 | | | | |
| Inn | Inn1 | 0.864 | 0.950 | 0.934 | 0.792 | Yes |
| | Inn2 | 0.878 | | | | |
| | Inn3 | 0.923 | | | | |
| | Inn4 | 0.912 | | | | |
| | Inn5 | 0.870 | | | | |
| Cus | Cus1 | 0.873 | 0.841 | 0.830 | 0.745 | Yes |
| | Cus2 | 0.865 | | | | |
| | Cus3 | 0.851 | | | | |

**Table 4.** *Cont.*

| Construct | Items | Outer Loading | Composite Reliability | Cronbach's Alpha | AVE | Discriminant Validity? |
|---|---|---|---|---|---|---|
| Res | Res1 | 0.803 | 0.936 | 0.917 | 0.708 | Yes |
| | Res2 | 0.873 | | | | |
| | Res3 | 0.854 | | | | |
| | Res4 | 0.868 | | | | |
| | Res5 | 0.806 | | | | |
| CompetAdv | CompetAdv1 | 0.818 | 0.926 | 0.901 | 0.716 | Yes |
| | CompetAdv2 | 0.818 | | | | |
| | CompetAdv3 | 0.850 | | | | |
| | CompetAdv4 | 0.859 | | | | |
| | CompetAdv5 | 0.883 | | | | |
| EcoS | EcoS1 | 0.902 | 0.935 | 0.897 | 0.829 | Yes |
| | EcoS2 | 0.931 | | | | |
| | EcoS3 | 0.897 | | | | |
| EnvS | EnvS1 | 0.885 | 0.918 | 0.866 | 0.788 | Yes |
| | EnvS2 | 0.865 | | | | |
| | EnvS3 | 0.912 | | | | |
| SocS | SocS1 | 0.902 | 0.943 | 0.910 | 0.838 | Yes |
| | SocS2 | 0.955 | | | | |
| | SocS3 | 0.904 | | | | |
| Moderating effect | CompAdv_EcoS_Res | 1.386 | 1.000 | 1.000 | 1.000 | Yes |

We verify that all indicators are credible because all item loadings are more significant than 0.773. Our data analysis shows that all the constructs are consistent because they are above 0.841. All AVEs are over 0.708 and demonstrate convergent validity. All Cronbach's Alpha readings for all structures in our analysis are more than 0.830.

*4.3. Structural Model Assessment*

To analyze the structural model's quality, we use a PLS and bootstrapping approach using 5000 subsamples [89]. Figure 2 summarizes our findings of the structural model. Table 5 demonstrates our hypotheses test results. These results indicate that our proposed hypotheses (Section 3) are all supported as justified below.

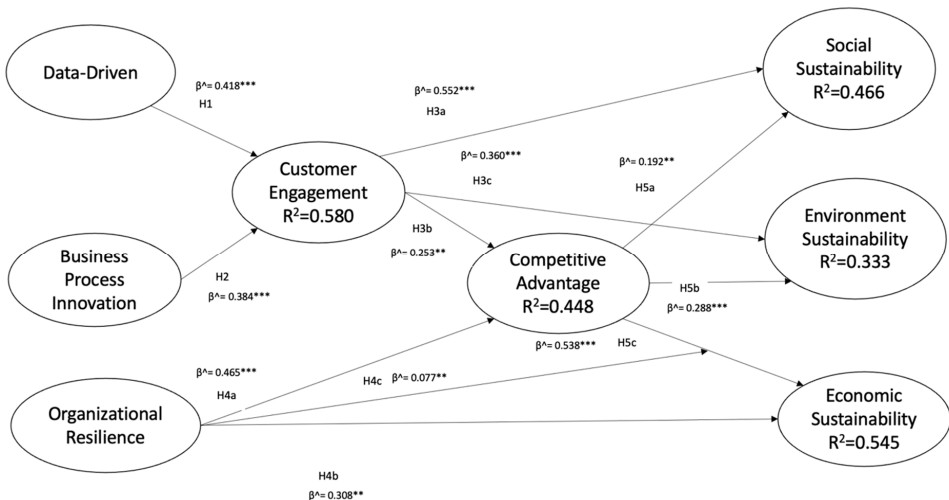

**Figure 2.** Sustainability explained by digital transformation model results. ** significant at $p < 0.01$; *** significant at $p < 0.001$.

**Table 5.** Hypothesis test results.

| Hypothesis | Independent Variable | Dependent Variable | Moderator | $F^2$ | Effect Size | *p*-Value | Findings | Conclusion |
|---|---|---|---|---|---|---|---|---|
| H1 | Data-Driven | Customer Engagement | - | 0.150 | Medium | 0.000 | Positively & Statistically Significant *** ($\hatβ$ = 0.418, $p < 0.001$) | Supported with large effect |
| H2 | Business Process Innovation | Customer Engagement | - | 0.127 | Small | 0.000 | Positively & Statistically Significant *** ($\hatβ$ = 0.384, $p < 0.001$) | Supported with large effect |
| H3a | Customer Engagement | Social Sustainability | - | 0.376 | Large | 0.000 | Positively & Statistically Significant *** ($\hatβ$ = 0.552, $p < 0.001$) | Supported with large effect |
| H3b | Customer Engagement | Competitive Advantage | - | 0.057 | Small | 0.004 | Positively & Statistically Significant ** ($\hatβ$ = 0.253, $p < 0.05$) | Supported with medium effect |
| H3c | Customer Engagement | Environmental Sustainability | - | 0.128 | Small | 0.000 | Positively & Statistically Significant *** ($\hatβ$ = 0.360, $p < 0.001$) | Supported with large effect |
| H4a | Organizational Resilience | Competitive Advantage | - | 0.193 | Medium | 0.000 | Positively & Statistically Significant *** ($\hatβ$ = 0.465, $p < 0.001$) | Supported with large effect |
| H4b | Organizational Resilience | Economic Sustainability | - | 0.120 | Small | 0.001 | Positively & Statistically Significant ** ($\hatβ$ = 0.308, $p < 0.05$) | Supported with medium effect |
| H4c | Organizational Resilience | Economic Sustainability | Competitive Advantage | 0.021 | Small | 0.030 | Positively & Statistically Significant ** ($\hatβ$ = 0.077, $p < 0.05$) | Supported with medium effect |
| H5a | Competitive Advantage | Social Sustainability | - | 0.046 | Small | 0.016 | Positively & Statistically Significant ** ($\hatβ$ = 0.192, $p < 0.05$) | Supported with medium effect |
| H5b | Competitive Advantage | Environmental Sustainability | - | 0.082 | Small | 0.000 | Positively & Statistically Significant *** ($\hatβ$ = 0.288, $p < 0.001$) | Supported with large effect |
| H5c | Competitive Advantage | Economic Sustainability | - | 0.356 | Large | 0.000 | Positively & Statistically Significant *** ($\hatβ$ = 0.538, $p < 0.001$) | Supported with large effect |

We start to explain this section by reporting the *p*-values and β^. H1 is supported with large effect because data-driven account for 41.8 percent of the variation in customer engagement (β^ = 0.418, *p* < 0.001). H2 is supported with large effect because business process innovation explains 38.4 percent of the variation in customer engagement (β^ = 0.384, *p* < 0.001). H3a is supported with large effect because customer engagement explains 55.2 percent of variation in social responsibility (β^ = 0.552, *p* < 0.001). H3b is supported with medium effect because customer engagement explains 25.3 percent of the variation in competitive advantage (β^= 0.253, *p* < 0.05). H3c is supported with large effect because customer engagement explains 36 percent of the variation in environmental sustainability (β^ = 0.360, *p* < 0.001). H4a is supported with large effect because organizational resilience explains 46.5 percent of the variance in competitive advantage (β^ = 0.465, *p* < 0.001. H4b is supported with medium effect, and organizational resilience accounts for 30.8 percent of the variation in economic sustainability (β^ = 0.308, *p* < 0.05). H4c is supported with medium effect because the moderating effect of organizational resilience explains 7.7 percent of variation in relation between competitive advantage and economic sustainability. H5a is supported with medium effect because competitive advantage explains 19.2 percent of the variation in social sustainability (β^ = 0.192, *p* < 0.05). H5b is supported with large effect because competitive advantage explains 28.8 percent of variation in environmental sustainability (β^ = 0.288, *p* < 0.001). H5c is supported with large effect because competitive advantage explains 53.8 percent of variation in economic sustainability (β^ = 0.538, *p* < 0.001).

We additionally report the $F^2$ indicator to determine if a concept is significant or not. This metric indicates a substantial influence if it is more than 0.350 ($F^2$ > 0.350), a modest influence if (0.350 > $F^2$ > 0.150), and a low influence if (0.150 > $F^2$ > 0.020) [90]. Our findings demonstrate that all the hypotheses are positive and significant, however their impact sizes vary. H3a, and H5c have large effects, H1, and H4a medium effects, and H2, H3b, H3c, H4b, H4c, H5a, and H5b small effects. In Table 5, you can find a summary of the consequences.

## 5. Discussion

This research applies digital transformation domains that were introduced by Rogers [5], and the theories of organizational resilience, disruption, business model, resource-based view, and platform, to propose a theoretical model for boosting organizational sustainability through digital transformation. In this model, social, environmental, and economic sustainability was measured by the effects of digital transformation's domain and organizational resilience.

Rogers [5] explains why businesses must reconsider their customers, data, innovation, and value and describes how they can use them, but he doesn't propose a model. Another study evaluates the direct effects of competition, customer, data, and innovation on sustainability, but does not consider their interactions with each other [91]. In contrast, we propose a theoretical model to study direct and indirect relations between these domains to sustainability. In addition, we add a new domain of organizational resilience and study its effect on sustainability. Moreover, we validate our theoretical model with our empirical study.

To propose the model, we design the constructs and their relationships based on some findings in previous research. Prior work shows that customer experience and innovation capabilities provide a competitive advantage and create value [48,52,56]. The other line of research shows that IT capabilities, digital platforms, and technological capabilities increase competitive advantages [5,31,34].

Our study shows that data-driven and business process innovation significantly affect customer engagement, where the impact of data-driven is higher than innovation. Moreover, the customer engagement significantly affects competitive advantage. Therefore, we discover an indirect effect of data and innovation on competitive advantage. Our results are aligned with findings in previous work.

In this research, we find that, the impact of competitive advantage on economic sustainability is greater than its effect on social and environmental sustainability. On the

other hand, competitive advantage effect is about 1.7 times greater than the impact of organizational resilience on economic sustainability. It validates the conclusions of earlier researchers who debated that competitive advantage and organizational resilience have an impact on sustainability [16,34,59,61,80,81].

Regarding customer perspective, a previous study presents a framework that demonstrates how customer engagement affects competitive advantage and company performance [54]. We also found that customer engagement has more impact on social and environmental sustainability than environmental sustainability. But these impacts are more than competitive advantage effects on social and environmental sustainability.

## 6. Conclusions and Implications

This study aims to understand the effect of digital transformation on sustainability. For this reason, the research model consists of data-driven, business process innovation, customer engagement, competitive advantage, organizational resilience, social sustainability, environmental sustainability, and economic sustainability. The research model explains 33.3% environmental sustainability, 54.5% economic sustainability, and 46.6% social sustainability. We found that competitive advantage has more impact on economic sustainability instead of social and environmental sustainability. Further, competitive advantage has more effect on economic sustainability than organizational resilience. In addition, customer engagement has more effect on social sustainability than environmental sustainability.

The study's theoretical implications add to the increasing knowledge by proposing a theoretical model that integrates digital transformation theory with organizational resilience and sustainability. We also conducted an empirical study to evaluate the research model to determine how sustainability is explained by digital transformation and organizational resilience.

There are also practical implications. The findings reveal customers' significant role in gaining valuable capabilities and competitive advantages in the organizations. Therefore, the companies should engage, attract, inspire, and collaborate with customers more in all aspects of organizations. For this reason, they can use valuable and meaningful data as strategic assets to engage more customers. They need to apply innovative solutions and technologies to reduce costs and improve quality to respond to customers' needs. Companies can involve their customers by gamification strategies to grant them rewards. For example, the customers of Starbucks can earn stars when they scan their barcode from the My Starbucks Reward app. On the other hand, competitive advantage will increase satisfaction with the company's performance in sales, profit, and cash flow. Also, competitive advantage impacts developing more environmentally friendly products that use fewer natural resources and decrease pollution. It can enhance a company's social recognition and empowerment in society. Therefore, the c-suite leaders can find competitive advantages for their companies by obtaining a set of traits that enable their firms to work better than their competitors to achieve more sustainability. Furthermore, suppose managers increase the resilience of their companies in response to the crisis and highlight the ability to recover and grow under uncertainty and emergency. In that case, they can achieve more competitive advantage and economic sustainability.

**Author Contributions:** Conceptualization, R.H., C.J.C. and M.A.; methodology, R.H., C.J.C. and M.A.; software, R.H., C.J.C. and M.A.; validation, R.H., C.J.C. and M.A.; formal analysis, R.H., C.J.C. and M.A.; investigation, R.H., C.J.C. and M.A.; resources, R.H., C.J.C. and M.A.; data curation, R.H., C.J.C. and M.A.; writing—original draft preparation, R.H., C.J.C. and M.A.; writing—review and editing, R.H., C.J.C. and M.A.; visualization, R.H., C.J.C. and M.A.; supervision, R.H., C.J.C. and M.A.; project administration, R.H., C.J.C. and M.A.; funding acquisition, C.J.C. All authors have read and agreed to the published version of the manuscript.

**Funding:** The authors acknowledge financial support via ADVANCE-CSG from the Fundação para a Ciência and Tecnologia (FCT Portugal) through research grant number UIDB/04521/2020, and we gratefully acknowledge the financial support from FCT—Fundação para a Ciencia e Tecnologia, I.P.,

(Portugal), national funding through research grant UIDB/04152/2020—Centro de Investigação em Gestão de Informação (MagIC).

**Institutional Review Board Statement:** The study was conducted according to the guidelines of the Declaration of Helsinki, and approved by the Ethics Committee of NOVA IMS (NFSYS2022-1-37926, on 1/3/2022).

**Informed Consent Statement:** Informed consent was obtained from all subjects involved in the study.

**Conflicts of Interest:** The authors declare no conflict of interest.

## Appendix A

**Table A1.** Measurement model.

| Construct | Measurement Items | Authors |
|---|---|---|
| Data-Driven | —Our data strategy is focused on how to turn data into new value.<br>—We manage our data as a strategic asset we are building over time.<br>—Our data is organized to be accessible by all divisions of the company. | [5] |
| Business process innovation | —Team leaders in our organization honor cutting-edge ideas for the innovation of business processes.<br>—Our top management rewards employees who present pioneering ideas for enhancing the performance of business processes.<br>—Our organization welcomes concepts for fundamental innovations that increase the performance of business processes.<br>—Our organization encourages thinking "outside the box" to create innovative solutions in business processes.<br>—Managers of our organization are open to radical changes that enhance the performance of business processes. | [92] |
| Customer Engagement | —We are focused on our customers' changing digital habits and path to purchase.<br>—We use marketing to attract, engage, inspire, and collaborate with customers.<br>—Our customers' advocacy is the biggest influence on our brand and reputation. | [93] |
| Organizational Resilience | —We have a focus on being able to respond to the unexpected.<br>—We proactively monitor our industry to have an early warning of emerging issues.<br>—We have clearly defined priorities for what is important during and after a crisis.<br>—There would be good leadership from within our organization if we were struck by a crisis.<br>—Our organization maintains sufficient resources to absorb some unexpected change.<br>—We can make tough decisions quickly. | [94] |
| Competitive Advantage | —Compared with competitors, the quality of the company's products or services is very good.<br>—Compared with competitors, the profitability of this company is very high.<br>—Compared with competitors, the company's product market share has grown rapidly.<br>—Compared with competitors, the company has a better reputation.<br>—Compared with competitors, our products are in an advantageous position in the market. | [95] |
| Economic Sustainability | —Please indicate your level of satisfaction with your company's performance in sales.<br>—Please indicate your level of satisfaction with your company's performance in Net Profit.<br>—Please indicate your level of satisfaction with your company's performance in Cash Flow. | [96] |
| Environmental Sustainability | —Compared with our major competitors, our products are more environmentally friendly.<br>—Compared with our major competitors, our production process requires fewer natural resources.<br>—Compared with our major competitors, our production process decreases environmental pollution. | [96] |
| Social Sustainability | —Our company enhances our social recognition in society.<br>—Our company improves our empowerment in society.<br>—Our company provides freedom and control. | [96] |

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
