# Peer review of "Boosting Sustainability through Digital Transformation’s Domains and Resilience"

_sustainability, doi:10.3390/su14031822_

Round 1

Reviewer 1 Report

I found the article interesting, since it focuses on two timely topics with academic interest. Generally, the article is well designed. When I started reviewing, I was clearly looking for well-crafted research questions. Following, I looked for a theoretical-conceptual model along with the definitions of the most relevant concepts/variables, and I was also able to find them. Moreover, I also looked for contributions to practice and theory that also seemed relevant to me. So, from a scientific point of view, I think the article is very well done. However, as a reader, I have at least one question that may be relevant to authors. For example, in line 328-330, `the generic arguments "companies should engage, attract, inspire and collaborate more with customers in all aspects of organizations" are used. I realize that in the next few lines there is some justification for this argument. But for junior researchers it can be difficult to materialize some of these aspects. Why no to present one or another case of real companies so that the reader can have a better perception? If is not ok to present it in the conclusions, at least throughout the text. This comment it's just an idea to improve the article, which the authors don't exactly need to follow. Overall, I think this is a good article.

Author Response

Ms. No.: sustainability-1557504

Title: Boosting Sustainability through Digital Transformation’s Domains and Resilience

Journal: [Sustainability] Manuscript ID: sustainability-1557504

Dear Anonymous Reviewer,

We would like to thank you for your valuable effort in reviewing our paper. Under your guidance, we have improved it to meet the high standards of the Research in Sustainability Journal. Please find below a list where we show how we addressed each suggestion and recommendation made in this round of revision. Also, to facilitate tracking, we have made “traction changes” to the original manuscript.

Comments

Response

1.     I found the article interesting, since it focuses on two timely topics with academic interest. Generally, the article is well designed. When I started reviewing, I was clearly looking for well-crafted research questions. Following, I looked for a theoretical-conceptual model along with the definitions of the most relevant concepts/variables, and I was also able to find them. Moreover, I also looked for contributions to practice and theory that also seemed relevant to me. So, from a scientific point of view, I think the article is very well done. However, as a reader, I have at least one question that may be relevant to authors. For example, in line 328-330, `the generic arguments "companies should engage, attract, inspire and collaborate more with customers in all aspects of organizations" are used. I realize that in the next few lines there is some justification for this argument. But for junior researchers it can be difficult to materialize some of these aspects. Why no to present one or another case of real companies so that the reader can have a better perception? If is not ok to present it in the conclusions, at least throughout the text. This comment it's just an idea to improve the article, which the authors don't exactly need to follow. Overall, I think this is a good article.

Thank you for your interest and for taking the time to review our paper. To address this concern, we provided an example in the conclusion, as follows:

“Therefore, the companies should engage, attract, inspire, and collaborate with customers more in all aspects of organizations. For this reason, they can use valuable and meaningful data as strategic assets to engage more customers. They need to apply innovative solutions and technologies to reduce costs and improve quality to respond to customers' needs. Companies can involve their customers by gamification strategies to get them rewards. For example, the customers of Starbucks can earn stars when they scan their barcode from the My Starbucks Reward app.”

.

Reviewer 2 Report

Dear Authors

It was a pleasure to read your article about a very interesting topic.

The paper is well done, well written and discussed.

Author Response

Ms. No.: sustainability-1557504

Title: Boosting Sustainability through Digital Transformation’s Domains and Resilience

Journal: [Sustainability] Manuscript ID: sustainability-1557504

Dear Anonymous Reviewer,

We would like to thank you for your valuable effort in reviewing our paper. Under your guidance, we have improved it to meet the high standards of the Research in Sustainability Journal. Please find below a list where we show how we addressed each suggestion and recommendation made in this round of revision. Also, to facilitate tracking, we have made “traction changes” to the original manuscript.

Comments

Response

It was a pleasure to read your article about a very interesting topic.

The paper is well done, well written and discussed.

Thank you for reviewing our study, and the resulting interest in our study. We carefully cleaned up our tables in the paper and improved the quality. We thank you for your comments and for taking the time to review the study. Your words have encouraged us to pursue this line of research.

Reviewer 3 Report

Interesting topic and research approach - discussion should be extended and other minor improvements are suggested

Author Response

Ms. No.: sustainability-1557504

Title: Boosting Sustainability through Digital Transformation’s Domains and Resilience

Journal: [Sustainability] Manuscript ID: sustainability-1557504

Dear Anonymous Reviewer,

We would like to thank you for your valuable effort in reviewing our paper. Under your guidance, we have improved it to meet the high standards of the Research in Sustainability Journal. Please find below a list where we show how we addressed each suggestion and recommendation made in this round of revision. Also, to facilitate tracking, we have made “traction changes” to the original manuscript.

Comments

Response

Interesting topic and research approach - discussion should be extended and other minor improvements are suggested

Thank you for reviewing our study, and the resulting interest in our study. We carefully cleaned up our tables in the paper and improved the quality. We thank you for your comments and for taking the time to review the study. Your words have encouraged us to pursue this line of research. We do minor improvements in abstract, introduction, and empirical results.
We also extended the discussion as you mentioned, as follows:
“This research applies digital transformation domains that introduced by Rogers [1], and the theories of organizational resilience, disruption, business model, resource-based view, and platform, to propose a theoretical model for boosting organizational sustainability through digital transformation. In this model, social, environmental, and economic sustainability was measured by the effects of digital transformation’s domain and organizational resilience.

Rogers [1] explains why businesses must reconsider their customers, data, innovation, and value and describes how they can use them, but he doesn’t propose a model. Another study evaluates the direct effects of competition, customer, data, and innovation on sustainability, but does not consider their interactions with each other [2]. In contrast, we propose a theoretical model to study direct and indirect relations between these domains to sustainability. In addition, we add a new domain of organizational resilience and study its effect on sustainability. Moreover, we validate our theoretical model with our empirical study.

 To propose the model, we design the constructs and their relationships based on some findings in previous research. Prior work shows that customer experience and innovation capabilities provide a competitive advantage and create value [3–5]. The other line of research shows that IT capabilities, digital platforms, and technological capabilities increase competitive advantages [1,6,7].

Our study shows that data-driven and business process innovation significantly affect customer engagement, where the impact of data-driven is higher than innovation. Moreover, the customer engagement significantly affects competitive advantage. Therefore, we discover an indirect effect of data and innovation on competitive advantage. Our results are aligned with findings in previous work.

In this research, we find that, the impact of competitive advantage on economic sustainability is greater that its effect on social and environmental sustainability. On the other hand, competitive advantage effect is about 1.7 times greater than the impact of organizational resilience on economic sustainability. It validates the conclusions of earlier researchers who debated that competitive advantage and organizational resilience have an impact on sustainability [6,8–12].

Regarding to customer perspective, previous study presents a framework that demonstrates how customer engagement affects competitive advantage and company performance [13]. We also found that customer engagement has more impact on social and environmental sustainability than environmental sustainability. But these impacts are more than competitive advantage effects on social and environmental sustainability.”

References:

  1. Rogers, D. The Digital Transformation Playbook: Rethink Your Business for the Digital Age; Columbia Business School Publishing, 2016;
  2. Hilali, W.E.; Manouar, A.E.; Janati Idrissi, M.A. Reaching Sustainability during a Digital Transformation: A PLS Approach. Int. J. Innov. Sci. 2020, 12, doi:DOI 10.1108/IJIS-08-2019-0083.
  3. Kunz, W.; Aksoy, L.; Bart, Y.; Heinonen, K.; Kabadayi, S.; Villaroel Ordenes, F.; Sigala, M.; Diaz, D.; Theodoulidis, B. Customer Engagement in a Big Data World. J. Serv. Mark. 2017, 31, 161–171, doi:DOI: 10.1108/JSM-10-2016-0352.
  4. Foroudi, P.; Jin, Zh.; Gupta, S.; Melewar, T.C.; Foroudi, M.M. Influence of Innovation Capability and Customer Experience on Reputation and Loyalty. J. Bus. Res. 2016, 69, 4882–4889, doi:https://doi.org/10.1016/j.jbusres.2016.04.047.
  5. Havir, D. Building Competetive Advantage through Customer Experience Management. Acta Acad. Karviniensia 2019, XIX, 28–41, doi:DOI: 10.25142/aak.2019.012.
  6. Lee, S.; Yoo, J. Determinants of a Firm’s Sustainable Competitive Advantages: Focused on Korean Small Enterprises. Sustainability 2021, 13, 346, doi:https://doi.org/10.3390/su13010346.
  7. Nwankpa, J.K.; Roumani, Y. IT Capability and Digital Transformation: A Firm Performance Perspective.; 2016.
  8. Berns, M.; Townend, A.; Khayat, Z.; Balagopal, B.; Reeves, M.; Hopkins, M.S.; Kruschwitz, N. Sustainability and Competitive Advantage. MIT Sloan Manag. Rev. 2009, 51.
  9. Kim, J.; Seok, B.; Choi, H.; Jung, S.; Yu, J. Sustainable Management Activities: A Study on the Relations between Technology Commercialization Capabilities, Sustainable Competitive Advantage, and Business Performance. Sustainability 2020, 12, 7913, doi:doi:10.3390/su12197913.
  10. Miceli, A.; Hagen, B.; Riccardi, M.P.; Sotti, F.; Settembre-Blundo, D. Thriving, Not Just Surviving in Changing Times: How Sustainability, Agility and Digitalization Intertwine with Organizational Resilience. Sustainability 2021, 13, 2052, doi:https://doi.org/10.3390/su13042052.
  11. Rai, S.S.; Rai, S.; Singh, N.K. Organizational Resilience and Social‑economic Sustainability: COVID‑19 Perspective. Env. Dev Sustain 2021, 23, 12006–12023, doi:https://doi.org/10.1007/s10668-020-01154-6.
  12. Srivastava, M.; Franklin, A.; Maritinette, L. Building a Sustainable Competitive Advantage. J. Technol. Manag. Innov. 2013, 8, doi:DOI: 10.4067/S0718-27242013000200004.
  13. Kumar, V.; Pansari, A. Competitive Advantage Through Engagement. J. Mark. Res. 2016, 53, 497–514.